# Non-Probabilistic Reliability Analysis of Robot Accuracy under Uncertain Joint Clearance

**Zhaoping Tang [1], Jun Peng [1], Jianping Sun [2],\* and Xin Meng [1]**

[1] School of Mechatronics and Vehicle Engineering, East China Jiaotong University, Nanchang 330013, China
[2] School of Transportation Engineering, East China Jiaotong University, Nanchang 330013, China
\* Correspondence: 1654@ecjtu.edu.cn

**Abstract:** The development of industrial robots in high-precision fields is currently constrained by the reliability of motion. Considering the influence of the joint clearance on the motion reliability of the industrial robot, the kinematic model of the industrial robot is established, the kinematic equation of the robot is deduced, and positive kinematic solutions are performed. The non-probability positioning accuracy reliability measure of a robot end-effector is proposed, based on the non-probability theory and method, combined with the prescribed permission interval and error interval, and different states of reliability can be judged according to the position relationship, the non-probability reliability properties are outlined, and the positioning accuracy reliability assessment model is established. Combined with the joint clearance modeling theory, the simulation of the robot end-effector under the influence of six joint clearances is carried out, and the displacement error interval of the end-effector under the preset motion path is analyzed for the industrial robot motion reliability problem. The motion path is split by time, and the end effector moves to different workspace areas in different time periods. The motion reliability of each segment is analyzed, and it is concluded that the reliability of the end-effector under the influence of uncertain joint clearance parameters changes in different working regions. Based on the above, the research direction of space division and partition parameter calibration is proposed, which lays a foundation for the study of partition non-probabilistic calibration of robot workspace.

**Keywords:** industrial robot; non-probability interval theory; joint clearance; motion reliability

## 1. Introduction

Industrial robots are extensively used in grinding, handling, loading, and unloading fields with the advantages of high efficiency and low cost, but their positioning accuracy is an important technical prerequisite to be applied in high-precision fields. The absolute positioning error of robots is mainly caused by the error of geometric parameters, which accounts for about 80%~90% of the absolute positioning error of industrial robots. It is important to explore the sources of absolute positioning error of robots and its influence and to study the reliability of robot motion accuracy under working conditions to enhance the accuracy of the position of the industrial robot [1].

In this regard, numerous scholars have conducted a lot of research work. For the uncertainty factors causing the motion accuracy error, Yu et al. took the crank slider mechanism as the object and analyzed the effect of full kinematic pair clearance on the mechanism dynamics characteristics [2]. Andrei et al. proposed that the reference positioning deviation existing in the structure of industrial robots would impact the motion accuracy of the robot end-effector, and the positioning deviation mainly consists of the vertical deviation of the joint and the parallelism deviation of the linkage [3]. Yan conducted an error analysis of the palletizing robot and discussed the effects of rod length error and angle error on the end position error, respectively [4]. To address the issue of motion accuracy reliability, Gu et al. analyzed the effects of the number of clearance joints and different clearance sizes on the

dynamic output characteristics of the floating-based space robot arm. They quantitatively analyzed the degree of influence of the clearance on the motion reliability of the floating-based space robot arm [5]. Huang et al. analyzed the motion accuracy of the 3-RRR parallel mechanism with consideration of the motion sub-clearance and component flexibility. It was proved that the clearance value was inversely proportional to the motion accuracy of the moving platform, and the error of the flexible member decreased compared to the rigid member [6]. Zhang et al. constructed a flexible robot dynamics model considering various uncertainties such as the more accurate structure of the robot and friction, the coupling properties of the joint and the flexible linkage, and analyzed the positioning accuracy of the flexible robot under the effect of various parameters [7]. Li et al. analyzed the reliability of the mechanism motion by considering the influence of dimensional error, universal joint clearance error, transmission error, and ball head clearance error on motion accuracy [8]. The above analysis mainly focuses on various error factors and investigates their effects on robot motion reliability, which provides an important role in the calibration compensation and absolute accuracy improvement of robots, but does not establish a relationship model between uncertainty variables and positioning accuracy of industrial robots, and lacks subsequent in-depth research on robot motion reliability.

In response to this situation, scholars have made many attempts to evaluate the reliability of positioning accuracy, improve calibration methods, and improve robot positioning accuracy. Wang et al. constructed the functional function of the robot system based on the probability distribution of the maximum entropy principle, and under a certain number of samples, the advantage of the fourth-order moment estimation method for a robot reliability solution was proved by comparing with the Monte Carlo method calculation [9]. Jiang et al. used a fuzzy set theory for the quantitative analysis of system reliability dynamically, expressed the failure probability of basic events with fuzzy triangular numbers, and assessed the robot system by both qualitative analysis and quantitative calculation [10]. Icli et al. proposed an automatic robot calibration method using coordinate measuring machines to obtain a large number of data coordinates and verified by comparison that this method is more accurate than the results of the measured values derived from the same use of a laser tracker [11]. Cui et al. proposed a method to use the Monte Carlo method for robot geometric parameter calibration to improve the reliability of industrial robot end accuracy [12]. Shang et al. established an error model considering dimensional error, universal joint clearance error, drive error, and ball head clearance error. Analyzed the effects of different errors on the kinematic accuracy, took drive error as the key factor to determine the reliability of kinematic accuracy and proposed an error compensation policy to control the drive angle, which can reduce the kinematic error and increase the kinematic accuracy of the robot [13]. Qi et al. divided the error into quantitative error and residual error, established an error model based on kinematics, used the Extended Kalman filter (EKF) algorithm to recognize the quantitative error, used the spatial interpolation algorithm to improve the approximate accuracy of the remaining errors, and proposed a new compensation method, which improved the absolute positioning accuracy by about 1 mm [14].

The above studies on the reliability of robot motion are rich and have good results in terms of calibration compensation and improvement of the absolute accuracy of robots. Among them, the studies on the reliability assessment of positioning accuracy are mostly analyzed by probabilistic methods, which rely on a large amount of raw data. In the actual working process of robots, it is usually more difficult to obtain the exact values of parameters, so the method of employing a non-probability interval theory to analyze the robot's reliability is gaining popularity among researchers. Sun et al. proposed a reliability measurement and analysis approach for robot positioning accuracy and parameter calibration based on a non-probability model [15]. Merlet proposed using interval analysis, introduced the basic theory of interval algorithms, and demonstrated that interval analysis could be used to deal with uncertainty in robots [16]. Lara-Molina et al. proposed a fuzzy kinematic reliability method to quantify the kinematic reliability index, which

considers the influence of kinematic constraints and clearances on the kinematic chain of the manipulator [17]. Carreras et al. evaluated different parameterization strategies to understand the approach to generating output distributions of failure probabilities by extending standard interval arithmetic with new abstractions called interval grids [18]. Wu et al. focused on the optimal allocation of joint tolerances with consideration of the positional and directional errors of the robot end effector and the manufacturing cost, used the interval analysis for predicting errors in the performance of robot manipulators, and studied the effects of the upper bounds on the minimum cost and relative deviations of the directional and positional errors of the end effector [19]. In summary, the interval method has more obvious advantages in processing data information when the value bounds are easier to determine, but the true distribution of probabilities of the data parameters is difficult to predict. Compared with the uncertainty expressions of the Monte Carlo method [20] and probabilistic fuzzy mathematics [21], which depend on a large amount of raw data, this paper proposes a non-probabilistic positioning accuracy reliability measure to analyze the robot motion reliability under the influence of uncertain joint clearance parameters. Due to the different expression formulas corresponding to different states of positioning accuracy reliability, it is complicated to establish an evaluation model of specific motion reliability, and the interval conditions corresponding to different reliability states need to be analyzed separately. This method can determine the reliability state and evaluate the reliability of the end-effector with fewer samples, which plays a simplifying role in the research methods and steps of reliability. For the industrial robot motion reliability problem, considering the principle of the influence of joint clearance on positioning accuracy, it is considered that the robot end-effector is affected by joint clearance to varying degrees in different workspace regions. Therefore, compared with the motion reliability studies of other authors mentioned previously, in this paper, the motion path is split by time, the end-effector moves to different workspace regions in different time periods, and the motion reliability of each section is analyzed.

## 2. Mathematical Modeling of Robotic Systems

### 2.1. Robot System Kinematics Positive Solution

The overall structure of the KR 5 arc industrial robot is built from aluminum alloy, which has strong rigidity and can be engaged in polishing and grinding applications. It also has a certain fixed load capacity and is the smallest arc welding robot from KUKA at present. Its structure mainly consists of the base, swivel base, large arm, small arm rod, small arm, wrist, and flange. The second axis adopts a front design, which effectively increases the effective working range of the robot, and all six joints are rotating pairs. All axes are driven by maintenance-free AC servo motors, and the control system drives each linkage to rotate and complete the production work tasks. In this paper, the robot motion model was built using SolidWorks software. In the experimental process, if the geometric parameters and joint vectors of each motion joint are known, the position and attitude of the end-effector can be found with respect to the reference coordinate system. The D-H (Denavit–Hartenberg) method is used to establish the robot's coordinate system, with the *z*-axis along the rotation axis of each rotating pair, and the established coordinate system for the linkage of each joint of the robot is shown in Figure 1.

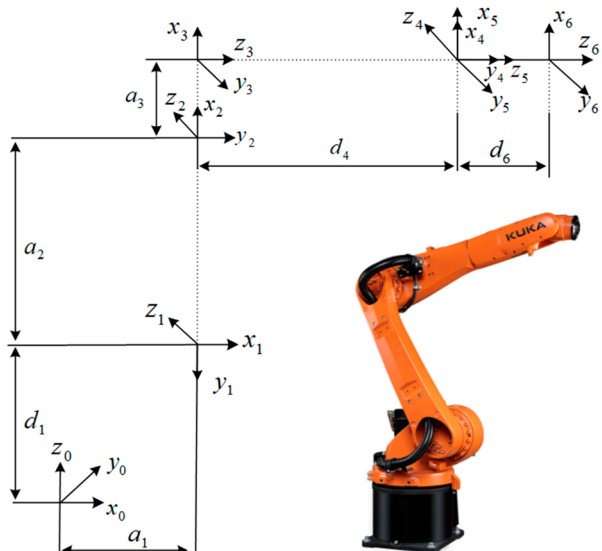

**Figure 1.** Robot linkage coordinate system.

When defining different poses of the robot, the transformation relationship between the coordinate system $i - 1$ and coordinate system $i$ can be described by rotational and translational motions, expressed as a chi-square transformation matrix:

$$A_i = {}_i^{i-1}T = Rot(z, \theta_i) Trans(0, 0, d_i) Trans(a_i, 0, 0) Rot(x, \alpha_i)$$
$$= \begin{bmatrix} \cos\theta_i & -\sin\theta_i\cos\alpha_i & \sin\theta_i\sin\alpha_i & a_i\cos\theta_i \\ \sin\theta_i & \cos\theta_i\cos\alpha_i & -\cos\theta_i\sin\alpha_i & a_i\sin\theta_i \\ 0 & \sin\alpha_i & \cos\alpha_i & d_i \\ 0 & 0 & 0 & 1 \end{bmatrix} \quad (1)$$

The theoretical parameters of the connecting rod are shown in Table 1. $a_i$ is the length of the connecting rod; $d_i$ is the offset distance of the connecting rod $i$ with respect to the connecting rod $i - 1$; $\alpha_i$ is the torsion angle of the connecting rod $i$; and $\theta_i$ is the joint rotation angle.

**Table 1.** Robot linkage and geometric parameters.

| $i$ | $a_i/(mm)$ | $d_i/(mm)$ | $\alpha_i/(°)$ | $\theta_i/(°)$ | Joint Range/$(°)$ |
|---|---|---|---|---|---|
| 1 | 180 | 400 | −90 | $\theta_1$ | (−170~170) |
| 2 | 600 | 0 | 0 | $\theta_2$ | (−180~65) |
| 3 | 40 | 0 | −90 | $\theta_3$ | (−120~165) |
| 4 | 0 | 620 | 90 | $\theta_4$ | (−180~180) |
| 5 | 0 | 0 | −90 | $\theta_5$ | (−120~120) |
| 6 | 0 | 80 | 0 | $\theta_6$ | (−360~360) |

Substituting the rod parameters in Table 1 into Equation (1), we can obtain the transformation matrix of each rod:

$$
{}_{1}^{0}T = \begin{bmatrix} \cos\theta_1 & 0 & -\sin\theta_1 & a_1\cos\theta_1 \\ \sin\theta_1 & 0 & \cos\theta_1 & a_1\sin\theta_1 \\ 0 & -1 & 0 & d_1 \\ 0 & 0 & 0 & 1 \end{bmatrix} \quad {}_{2}^{1}T = \begin{bmatrix} \cos\theta_2 & -\sin\theta_2 & 0 & a_2\cos\theta_2 \\ \sin\theta_2 & \cos\theta_2 & 0 & a_2\sin\theta_2 \\ 0 & 0 & 1 & d_2 \\ 0 & 0 & 0 & 1 \end{bmatrix}
$$

$$
{}_{3}^{2}T = \begin{bmatrix} \cos\theta_3 & 0 & -\sin\theta_3 & a_3\cos\theta_3 \\ \sin\theta_3 & 0 & \cos\theta_3 & a_3\sin\theta_3 \\ 0 & -1 & 0 & d_3 \\ 0 & 0 & 0 & 1 \end{bmatrix} \quad {}_{4}^{3}T = \begin{bmatrix} \cos\theta_4 & 0 & \sin\theta_4 & a_4\cos\theta_4 \\ \sin\theta_4 & 0 & -\cos\theta_4 & a_4\sin\theta_4 \\ 0 & 1 & 0 & d_4 \\ 0 & 0 & 0 & 1 \end{bmatrix} \quad (2)
$$

$$
{}_{5}^{4}T = \begin{bmatrix} \cos\theta_5 & 0 & -\sin\theta_5 & a_5\cos\theta_5 \\ \sin\theta_5 & 0 & \cos\theta_5 & a_5\sin\theta_5 \\ 0 & -1 & 0 & d_5 \\ 0 & 0 & 0 & 1 \end{bmatrix} \quad {}_{6}^{5}T = \begin{bmatrix} \cos\theta_6 & -\sin\theta_6 & 0 & a_6\cos\theta_6 \\ \sin\theta_6 & \cos\theta_6 & 0 & a_6\sin\theta_6 \\ 0 & 0 & 1 & d_6 \\ 0 & 0 & 0 & 1 \end{bmatrix}
$$

From the above equation, the homogeneous transformation matrix between the coordinate system on the robot end-effector and the reference coordinate system can be deduced as:

$$
{}_{6}^{0}T = {}_{1}^{0}T{}_{2}^{1}T{}_{3}^{2}T{}_{4}^{3}T{}_{5}^{4}T{}_{6}^{5}T \begin{bmatrix} n_x & o_x & a_x & p_x \\ n_y & o_y & a_y & p_y \\ n_z & o_z & a_z & p_z \\ 0 & 0 & 0 & 1 \end{bmatrix} \quad (3)
$$

where $(n_x, n_y, n_z)^T$, $(o_x, o_y, o_z)^T$, and $(a_x, a_y, a_z)^T$ are the unit direction vectors of the *x*-axis, *y*-axis, and *z*-axis of the robot end coordinate system under the robot base coordinate system, respectively; and $(p_x, p_y, p_z)^T$ are the position vectors of the robot end coordinate system origin under the robot base coordinate system. This equation is the positive solution of industrial robot kinematics.

$$
\begin{aligned}
n_x &= c_1(c_{23}(c_4c_5c_6 - s_4s_6) - s_{23}s_5c_6) + s_1(s_4c_5c_6 + c_4s_6) \\
n_y &= s_1(c_{23}(c_4c_5c_6 - s_4s_6) - s_{23}s_5c_6) - c_1(s_4c_5c_6 + c_4s_6) \\
n_z &= -s_{23}(c_4c_5c_6 - s_4s_6) - c_{23}s_5c_6 \\
o_x &= c_1(c_{23}(-c_4c_5s_6 - s_4c_6) + s_{23}s_5s_6) + s_1(-s_4c_5s_6 + c_4c_6) \\
o_y &= s_1(c_{23}(-c_4c_5s_6 - s_4c_6) + s_{23}s_5s_6) - c_1(-s_4c_5s_6 + c_4c_6) \\
o_z &= s_{23}(c_4c_5s_6 + s_4c_6) + c_{23}s_5s_6 \\
a_x &= -c_1(c_{23}c_4s_5 + s_{23}c_5) - s_1s_4s_5 \\
a_y &= -s_1(c_{23}c_4s_5 + s_{23}c_5) + c_1s_4s_5 \\
a_z &= s_{23}c_4s_5 - c_{23}c_5 \\
p_x &= c_1(c_{23}(a_3 - c_4s_5d_6) - s_{23}(c_5d_6 + d_4) + c_2a_2 + a_1) - s_1s_4s_5d_6 \\
p_y &= s_1(c_{23}(a_3 - c_4s_5d_6) - s_{23}(c_5d_6 + d_4) + c_2a_2 + a_1) + c_1s_4s_5d_6 \\
p_z &= s_{23}(c_4s_5d_6 - a_3) - c_{23}(c_5d_6 + d_4) - s_2a_2 + d_1
\end{aligned} \quad (4)
$$

Here: *s* represents $\sin(\cdot)$, *c* represents $\cos(\cdot)$, and its subscript number *i* corresponds to joint angle $\theta_i$, and when the subscript number is *ij*, it represents $\theta_i + \theta_j$. For example, $c_1 = \cos\theta_1$, $s_{23} = \sin(\theta_2 + \theta_3)$, and $c_{23} = \cos(\theta_2 + \theta_3)$.

The values of the six joint parameters can be used to calculate the end position of the robot, the robot kinematic modeling and simulation are carried out using the robot toolbox in MATLAB software, and the results of the robot kinematic simulation are shown in Figure 2.

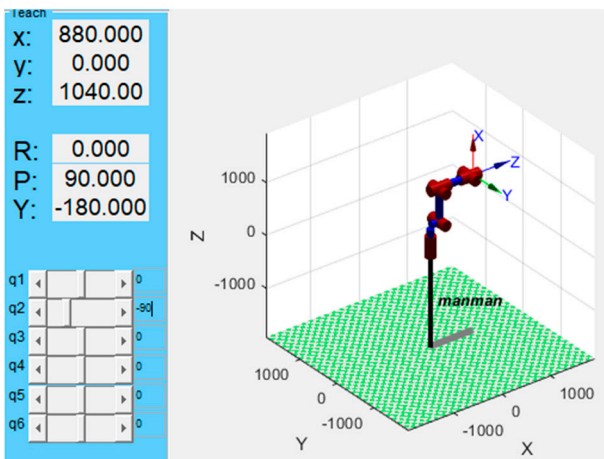

**Figure 2.** Robot kinematics simulation results.

Set the six joint rotation angles of a group of robots. Through the robot kinematics Equation (3) derived above, the robot end-effector's pose matrix can be solved in MATLAB, and then input the same joint rotation angle value in the simulation teaching interface shown in Figure 2 to obtain the end position coordinates. It can be seen that the calculation results agree with the simulation results by comparing the two groups of end position coordinates, and the correctness of D-H parameter selection and homogeneous transformation matrix calculation is verified.

### 2.2. Reliability Model for Robot Motion Based on Non-Probability Interval Theory

Interval mathematics expresses variables in terms of interval numbers, and the interval range obtained by interval operations contains all possible values of the calculation result. For any uncertain parameter $x$, whose value varies within a certain interval range, then $x$ can be expressed as $x^I = \left[ x^l, x^u \right] (x \in R, x^l \leq x_i \leq x^u)$, where $x^l$ and $x^u$ are two real numbers in the real number field $R$, and $x^l \leq x^u$, $x^l$ and $x^u$ are used as the lower and upper bounds of the range of values of $x$, respectively [22]. In particular, if the lower bound of $x^I$ is equal to the upper bound ($x^l = x^u$), then $x^I$ is said to degenerate to a real number, and any result derived from $x^I$ is also deterministically real, $x^I = \left[ x^l, x^u \right] = R$.

The four operations of interval mathematics are [23,24]:

$$
\begin{cases}
x + y \in \left[ x^l + y^l, x^u + y^u \right] \\
x - y \in \left[ x^l - y^u, x^u - y^l \right] \\
x \cdot y = \begin{cases} [0,0], x \in [0,0] \text{ or } y \in [0,0] \\ \left[ \min\left\{ x^l \cdot y^l, x^l \cdot y^u, x^u \cdot y^l, x^u \cdot y^u \right\}, \max\left\{ x^l \cdot y^l, x^l \cdot y^u, x^u \cdot y^l, x^u \cdot y^u \right\} \right], \text{ other} \end{cases} \\
x/y = \begin{cases} \left[ x^l, x^u \right] \cdot \left[ 1/y^u, 1/y^l \right], 0 \notin \left[ y^l, y^u \right] \\ [0,0], x \in [0,0] \\ \left[ \min\left\{ x^l/y^u, x^l/y^l, x^u/y^l, x^u/y^u \right\}, \max\left\{ x^l/y^l, x^l/y^u, x^u/y^l, x^u/y^u \right\} \right], \text{ other} \end{cases}
\end{cases}
\tag{5}
$$

Considering the response of the robot end position error to the input parameters, let the permissible interval of the robot end position error be known as:

$$
r \in r^I = [r^l, r^u]
\tag{6}
$$

Then, let the actual error function of the robot's end position be:

$$
s(\mathbf{x}) = s(x_1, x_2, \cdots, x_n)
\tag{7}
$$

Then, the functional equation of the robot's end position error is:

$$M = g(\mathbf{x}, r) = r - s(\mathbf{x}) = r - s(x_1, x_2, \cdots, x_n) \tag{8}$$

Here, $\mathbf{x} = (x_1, x_2, \cdots, x_n)$ $(x_i \in x_i^I, i = 1, 2, \cdots, n)$ is the input vector affecting the robot end position error response (including kinematic parameter variables, such as linkage torsion angle, linkage length, linkage offset distance, joint rotation angle, joint torsion angle, etc.). When $s(x_1, x_2, \cdots, x_n)$ is continuous about $x_i$ $(i = 1, 2, \cdots, n)$, $s(\mathbf{x})$ also belongs to a certain interval, and let $s^u$ and $s^l$ be the upper and lower bounds of the values taken by $s(\mathbf{x})$, respectively, then:

$$
\begin{aligned}
M = g(\mathbf{x}, r) &= r - s(\mathbf{x}) = r - s(x_1, x_2, \cdots, x_n) \\
&= r^I - s^I = [r^l - s^u, r^u - s^l]
\end{aligned} \tag{9}
$$

Obviously, $M$ is an interval number. Let $M^l = r^l - s^u$, $M^u = r^u - s^l$, then $M$ can be expressed as:

$$M \in M^I = [M^l, M^u] \tag{10}$$

From the definition of the functional equation of the robot end position error (Equation (8)), it can be seen that the robot end position error must satisfy the requirement when $M = g(\mathbf{x}, r) > 0$ (both $M^l$ and $M^u$ are greater than 0); the robot end position error must not satisfy the requirement when $M = g(\mathbf{x}, r) < 0$ ($M^l$ and $M^u$ are both less than 0); the robot end position error is in the critical state of meeting and not meeting the requirements when $M = g(\mathbf{x}, r) = 0$ ($M^l$ and $M^u$ are equal to 0). Additionally, when $M^l < 0$ and $M^u > 0$, part of the value of $M = g(\mathbf{x}, r)$ may be greater than 0, while the other part may be less than 0. At this time, the robot end position error does not necessarily meet the requirements. This corresponds to the case in Figure 3 below, respectively.

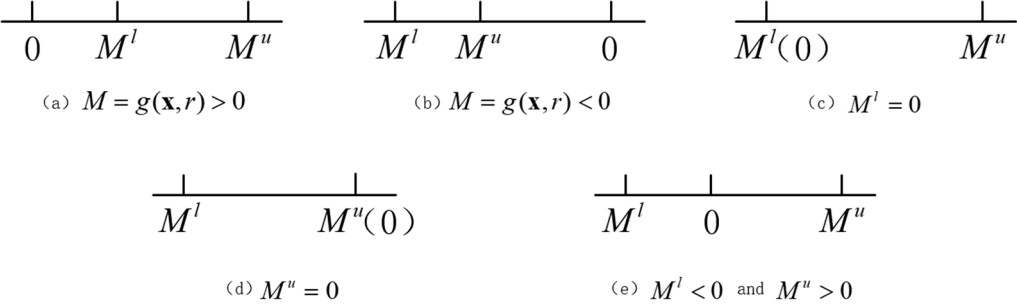

**Figure 3.** Interval function.

According to the above analysis, the robot end position reliability can be measured by comparing the value of the function $M$ of the robot end position error expressed by the interval number with the number 0. In addition, using non-probability reliability [25] (the ratio of the area of the reliability domain to the total area of the domain of the basic interval variables) can be a more convenient measure of the reliability of the robot end position.

Figure 4 shows the relationship between the position error interval and the allowable error interval on the coordinate system. Let $\psi = \left\{ (r, s) \middle| r \in \left[ r^l, r^u \right], s \in \left[ s^l, s^u \right] \right\}$ be the overlap area between the position error interval and the allowable error interval, $\Omega_R = \{(r, s) | M > 0\}$ be the reliability domain, $\Omega_F = \{(r, s) | M < 0\}$ be the failure domain, and the critical plane be $r^I - s^I = 0$. Then, the non-probability reliability of the robot end position is defined as:

$$R_{set} = \psi_f / \psi \tag{11}$$

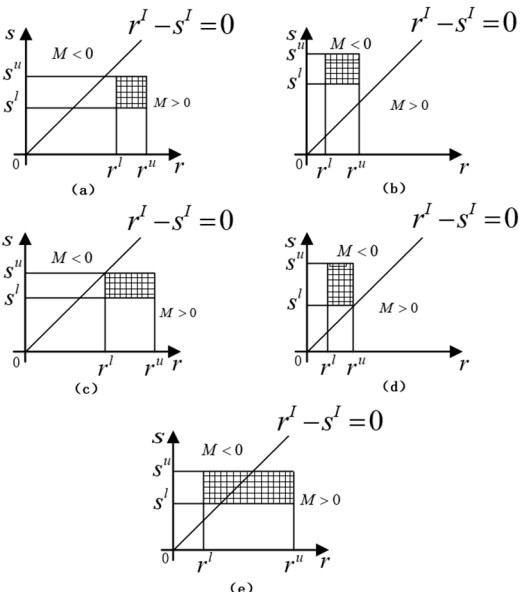

**Figure 4.** Interference model for robot end-effector position error and allowable accuracy.

Here: $\psi$ denotes the total area of the rectangle enclosed by the lines $r = r^l$, $r = r^u$, $s = s^l$, and $s = s^u$. $\psi_f$ denotes the part of the region $\psi$ overlapping with the reliability domain $\Omega_R$.

Obviously, the reliability $R_{set}$ of the robot's end position is a real number. By introducing the non-probability reliability $R_{set}$, the reliability of the robot end position can be measured more easily. It can be seen that the larger $R_{set}$ is, the higher the reliability of the robot's end position.

Obviously, in Figure 4a, $\psi \subset \Omega_R$, the positioning accuracy of the robot end-effector meets the requirements, and it is in a reliable state; in Figure 4b, $\psi \subset \Omega_F$, the positioning accuracy of the robot end-effector does not meet the requirements and is in a failure state; in Figure 4c, $\psi \subseteq \Omega_R$ and $\psi \cap \{(r,s)|M=0\} = \{r^l, s^u\}$, the positioning accuracy of the robot end-effector is in a reliable critical state; in Figure 4d, and $\psi \cap \{(r,s)|M=0\} = \{s^l, r^u\}$, the positioning accuracy of the robot end-effector is in a failure critical state; and in Figure 4e, $\psi \cap \Omega_F \neq \phi$ and $\psi \cap \Omega_R \neq \phi$, $r^l < s^l < s^u < r^u$, the positioning accuracy of the robot end-effector is in an unstable state. In this state, the size relationship of $r^u$, $s^l$, $s^u$, and $r^u$ affects the shape of the region $\psi$ where the position error interval and the allowable error interval overlap each other on the coordinate system, but both intersect with the critical plane $r^I - s^I = 0$. By analyzing the coordinate system, the reliability is less than 0.5 if $r^l < s^l < r^u < s^u$ for an unstable condition and more than 0.5 if $s^l < r^l < s^u < r^u$ for a reliable condition.

It can be seen that the non-probabilistic reliability has the following properties:

$$R_{set}(M(r,s) > 0) = \begin{cases} 1, & r^l \geq s^u \\ 0 < \frac{\psi_f}{\psi} < 1, & r^l > s^u \\ 0, & r^u < s^l \\ 0.5, & r^c = s^c \\ > 0.5, & r^c > s^c \\ < 0.5, & r^c < s^c \end{cases} \qquad (12)$$

Here: $r^c$, and $s^c$ are the midpoints of $R$ and $S$, respectively. $r^c = (r^l + r^u)/2, s^c = (s^l + s^u)/2$.

Considering that the overlap condition (overlap or not and the degree of overlap) between the actual error interval of robot position and the error permit value interval determines the reliability of positioning accuracy, based on the concept of structural safety proposed by Lingling Li et al. [25], the non-probability reliability of robot positioning accuracy $R_{set}$ is defined as the ratio of the length of the actual error interval that does not exceed the error permit value (length of the safety interval) to the length of the total variable domain of the basic interval [15]. The derivation of Figure 4 and equation (11) finally leads to a reliability model for the position accuracy of the robot end-effector as:

$$R_{\text{S}et} = 1 - \min\left\{1, \frac{(r^u \leq s^u) \times (s^u - r^u) + (r^l \geq s^l) \times (r^l - s^l)}{s^u - s^l}\right\} \tag{13}$$

Here: The result is 1 when $r^u \leq s^u$ and $r^l \geq s^l$ holds and 0 otherwise. $\min\left\{1, \frac{(r^u \leq s^u) \times (s^u - r^u) + (r^l \geq s^l) \times (r^l - s^l)}{s^u - s^l}\right\}$ indicates a minimum value between 1 and $\frac{(r^u \leq s^u) \times (s^u - r^u) + (r^l \geq s^l) \times (r^l - s^l)}{s^u - s^l}$ to ensure that the ratio of the length not exceeding the error allowance value to the total length of the domain of the underlying interval variable is not greater than 1.

$R_{set} \in [0, 1]$, When $R_{set} = 1$, the position accuracy is absolutely reliable; when $R_{set} = 0$, the position accuracy is unreliable; when $R_{set}$ is closer to 1, the position accuracy is more reliable, and vice versa. During robot motion, the position of the end-effector changes continuously, generating different position errors, and the reliability of the robot's positioning accuracy changes continuously. The reliability of the robot during its motion can be obtained by ignoring the cumulative errors in its motion.

## 3. Parametric Modeling of 6-Degree-of-Freedom Robots

The 3D model of the 6-degree-of-freedom robot was built using SolidWorks software and saved in Parasolid format. In the Adams environment, the mass, material properties, kinematic pair-constraints, and drive functions are added to each joint of the parametric model, respectively, and the simulation time is set to 10s, and the simulation step size is 0.1. Finally, the dynamics simulation of the established industrial robot model is carried out to analyze its motion reliability.

### 3.1. Joint Clearance Modeling Ideas

Generally speaking, the existence of joint clearance is due to manufacturing errors in the production of components, installation errors between components, and wear and tear between components during operation, so the magnitude of joint clearance is uncertain. This uncertainty in the amount of clearance will definitely lead to instability in the motion and dynamic performance of the industrial robot during operation. Therefore, to study the influence of the uncertainty of the joint clearance on the reliability of the positioning accuracy of industrial robots, the first step is to analyze the joint clearance.

Figure 5 shows the vector diagram containing the joint clearance, $p$ is the center of the hole; $c$ is the center of the shaft, $x$ and $y$ are the relative coordinates of the shaft center $c$, and the relative coordinates take $p$ as the origin, and the $op$ direction as the $x$-positive direction; the shaft moves inside the hole, and under the influence of the joint clearance, its $c$ and $p$ points do not coincide, and the center of the shaft is randomly distributed within the error circle, whose radius of the error circle is the difference between the radius of the hole and the shaft; $\vec{a}$ is the joint clearance vector, so the ideal length $l$ is no longer accurate at this time and $oc$ is the length of the member that actually contains the joint clearance, set to $L$, then:

$$L = \sqrt{(l + x)^2 + y^2} \tag{14}$$

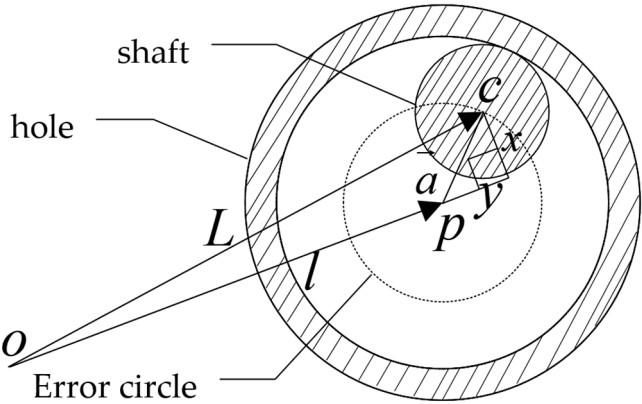

**Figure 5.** Joint clearance vector image.

$c$ always moves in the error circle with $p$ as the center of the circle, can obtain:

$$x^2 + y^2 \leq |a|^2 \tag{15}$$

Under the influence of the joint clearance, the length of $pc$ cannot be neglected, and since the center of the shaft $c$ is randomly distributed within the error circle, the clearance vector $\vec{a}$ is not the same for each joint, and the randomness of the joint clearance will certainly lead to the randomness of the end-effector point.

Figure 6a shows a diagram of the joints of the KR 5 arc industrial robot, which shows that the robot is a spatial 6 degrees of freedom rotating articulated robot. Figure 6b shows a schematic diagram of the robot joint clearance. Under the action of the six joint clearances, the actual spatial position $p_1$ of the robot end-effector is significantly shifted compared to the ideal spatial position $p_0$. Let the spatially distributed joint clearance vectors are $s_1$, $s_2$, $s_3$, $s_4$, $s_5$ and $s_6$, respectively, can obtain:

$$l_{p_0 p_1} = s_1 + s_2 + s_3 + s_4 + s_5 + s_6 \tag{16}$$

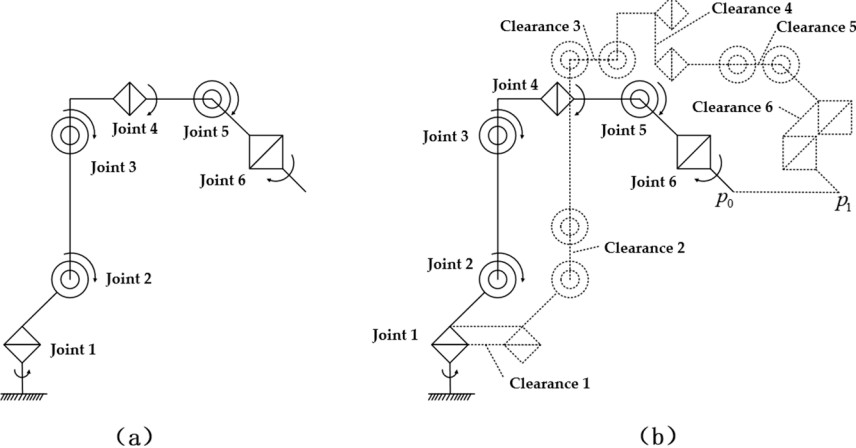

(a)　　　　　　　　　　　　　　　　　　　(b)

**Figure 6.** Diagram of the joint clearance of the robot mechanism.

By spatially synthesizing the joint clearance vector, the position error of the robot end-effector can be finally gained, and then the relationship between the joint clearance and the position accuracy of the 6R industrial robot end-effector can be established. Parametric modeling and simulation of the joint clearance can analyze the degree of influence on the reliability of the positioning accuracy of the robot end-effector under the joint clearance error.

### 3.2. Parametric Modeling of Joint Clearance

We create circles and cylinders at the model joint joints that are co-axial with the parts, respectively, the radius is represented by the design variable DV, and then the Boolean operation of Adams is used to merge them with the part, thus realizing the parametric modeling of the part. As a result, the interaction of the two rings and cylinders with different radii at each joint can be regarded as the fit between the two connected parts, and the joint clearance is represented as the fit clearance between the shaft and the hole. As shown in Figure 7, the model has a total of six joints, so 12 parametric axes and holes need to be created.

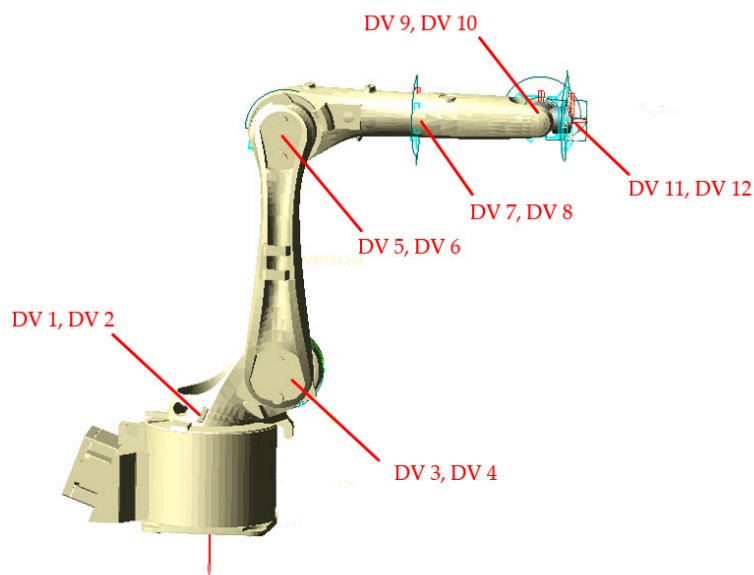

**Figure 7.** Diagram of parameter setting at each joint.

After establishing a parametric model of the six sets of joint clearances, the variables were designed for each joint axis hole. The H6/h5 criterion of the basic bore system was selected as follows in Table 2 below to set the specific values of the variables in Adams [26].

**Table 2.** Table of values of design variables for shafts and holes at joints.

| Variable | Nominal Size /(mm) | Tolerance /(mm) | Upper and Lower Limit Dimensions /(mm) | Maximum Clearance /(mm) | Minimum Clearance /(mm) | Notes |
|---|---|---|---|---|---|---|
| DV 1 | 40 | 0.011 | 39.989,40 | 0.027 | 0 | Radius of base shaft |
| DV 2 | 40 | 0.016 | 40,40.016 | | | Radius of swivel base hole |
| DV 3 | 30 | 0.013 | 30,30.013 | 0.022 | 0 | Radius of swivel base hole |
| DV 4 | 30 | 0.009 | 29.991,30 | | | Radius of large arm shaft (front end) |
| DV 5 | 22.5 | 0.009 | 22.491,22.5 | 0.022 | 0 | Radius of large arm shaft (end) |
| DV 6 | 22.5 | 0.013 | 22.5,22.513 | | | Radius of small arm rod hole |
| DV 7 | 10 | 0.006 | 9.994,10 | 0.015 | 0 | Radius of small arm rod shaft |
| DV 8 | 10 | 0.009 | 10,10.009 | | | Radius of small armhole (front end) |
| DV 9 | 6 | 0.008 | 6,6.008 | 0.013 | 0 | Radius of small armhole (end) |
| DV 10 | 6 | 0.005 | 5.995,6 | | | Radius of wrist shaft |
| DV 11 | 5 | 0.008 | 5,5.008 | 0.013 | 0 | Radius of wrist hole |
| DV 12 | 5 | 0.005 | 4.995,5 | | | Radius of flange shaft |

The change range of the joint turning angle is determined, and the limit position of the variation range enables the overall working space of the industrial robot to be determined. All work tasks and simulations are carried out in the workspace.

### 3.3. Setting of Model Drive Parameters

There should be some constraint relationship between the parts of the robot. One component restricts the motion of another component to ensure that there is correct motion between the individual components. Therefore, it is first necessary to preprocess the model for simulation. The holes and axes parameterized at each joint are connected by a rotating pair. In order to limit the joint deflection in the axial direction in the simulation, a plane pair is applied between the mating shafts and holes, as shown in Figure 8, to ensure that the rotation at the joints does not deviate from the plane of rotation.

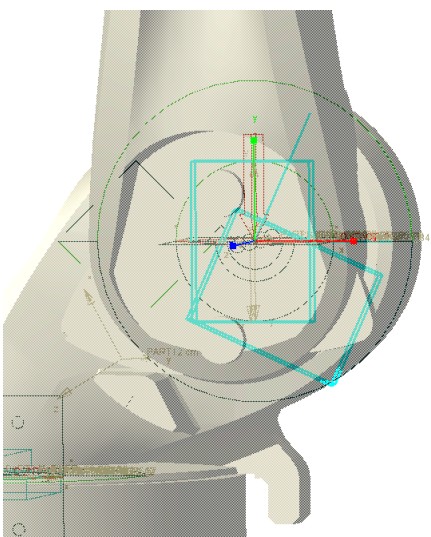

**Figure 8.** Setting of Adams-based joint constraints.

Considering that frictional impacts occur in the actual process due to the existence of clearance, contact forces are applied between the parameterized axis holes to simulate the clearance contact during operation, and the point drive method is used to drive the industrial robot's motion. The motion constraints and drives of the main moving components of each part of the model are set in Table 3 below.

**Table 3.** Diagram of each joint constraint and drive parameter setting.

| Joint | Connected Components | Constraints (Plane Pairs) | Contact Force | Driving Mode (Point Driving) | Drive Function |
|-------|----------------------|---------------------------|---------------|------------------------------|----------------|
| Joint 1 | Base, swivel base | joint_1 | contact_1 | general_motion_1 | STEP(time, 0, 0 d,1, −45 d) + STEP(time, 5, 0 d, 6, 45 d) |
| Joint 2 | Swivel base, large arm | joint_2 | contact_2 | general_motion_2 | STEP(time, 1, 0 d,2 ,60 d) + STEP(time, 6, 0 d, 7, −60 d) |
| Joint 3 | Large arm, small arm rod | joint_3 | contact_3 | general_motion_3 | STEP(time, 2, 0 d, 3, 60 d) + STEP(time, 7, 0 d, 8, −60 d) |
| Joint 4 | Small arm rod, small arm | joint_4 | contact_4 | general_motion_4 | STEP(time, 3, 0 d, 4, 90 d) + STEP(time, 8, 0 d, 9, −90 d) |
| Joint 5 | Small arm, wrist | joint_5 | contact_5 | general_motion_5 | STEP(time, 4, 0 d, 5, 45 d) + STEP(time, 9, 0 d, 10, −45 d) |
| Joint 6 | Wrist, flange | joint_6 | contact_6 | general_motion_6 | STEP(time, 4, 0 d, 5, 180 d) + STEP(time, 9, 0 d, 10, −180 d) |

## 4. Simulation Analysis of Positioning Accuracy of Industrial Robots

In Adams, we simulate the position and trajectory of the end-effector coordinates of the industrial robot based on the parameters set, derive the error range in each direction of the simulated data, and perform a reliability analysis based on non-probability interval theory to study the variation of robot kinematic reliability and analyze its motion reliability.

### 4.1. Analysis of End-Effector Coordinate Points

Referring to the structure and joint rotation range of the KR 5 arc industrial robot, a set of joint angles is selected in the reasonable activity space of each joint of the industrial robot. The end effector spatial position corresponding to this joint angle state is solved by using the robot positive kinematic formula given in Chapter 2, which proves to be basically consistent with the simulation coordinate position of the built model. The clearance variable is activated, and the simulation is repeated 100 times through the batch program to obtain the simulation data of the robot end-effector in three directions, as shown in Figure 9 for the distribution of the robot end-effector in spatial coordinates under the influence of the six joint clearances.

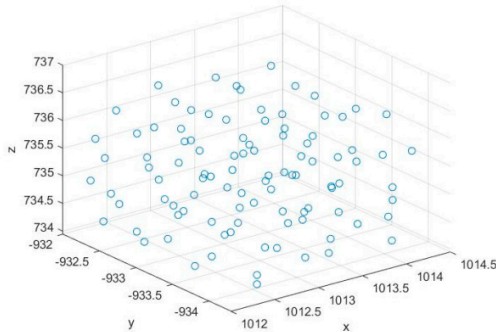

**Figure 9.** Scatter diagram of robot spatial coordinates.

From Figure 9, it can be seen that the robot end-effector coordinates have an error of no more than 2.5 mm in the x and y directions and no more than 3 mm in the z-direction, and the specific data of the end-effector in the x, y, and z directions, respectively, are derived through the powerful function of the Adams/PostProcessor module, and then the deviation interval of each coordinate position is obtained. As shown in Table 4, using equation (13) to calculate and solve the reliability of positioning accuracy in each direction, and 9–11 columns to solve the reliability by the traditional probability analysis method [27] (assuming normal distribution and taking the displacement allowable output limit error value eigenvalue: $(\mu_0, \sigma_0{}^2) = (0.8, 0.071111)$), it can be seen that the actual spatial coordinates of the robot end-effectors are more reliable than 0.9 in all three directions, which basically meets the requirements of positioning accuracy reliability.

**Table 4.** Comparison data of positioning accuracy and reliability.

| Joint Angle /(°) | Theoretical Coordinates | Range of Error /(mm) | | | Non-Probability Reliability | | | Traditional Probabilistic Reliability | | |
|---|---|---|---|---|---|---|---|---|---|---|
| | | $\Delta x$ | $\Delta y$ | $\Delta z$ | x | y | z | x | y | z |
| (−45°, −30°, −60°, −90°, −45°, 180°) | 1013.1, −933.109, 735.25 | −1.1424 | −1.077 | −1.0096 | | | | | | |
| | | | | | 0.909 | 0.9435 | 0.925 | 0.903 | 0.924 | 0.919 |
| | | 1.0588 | 1.0428 | 1.1521 | | | | | | |

By comparing the data of "Non-probability reliability" and "Traditional Probabilistic Reliability" in Table 4, we can find that the two methods are consistent in evaluating the robot's end However, the non-probability reliability method is less demanding, so it is more advantageous for the analysis of reliability problems of small sample and poor information structure systems. Under the consideration of the uncertain influence of joint clearance, the introduction of non-probability reliability theory to analyze the robot's accuracy has a wide application prospect.

### 4.2. Analysis of End-Effector Trajectories

According to the joint constraints and drive set for the robot in the previous paper, the motion trajectory simulation of the end-effector is carried out. The motion trajectory is shown in Figure 10. The joints of the robot make the rotational motion, and the motion trajectory of the end-effector is circular in the XY plane and is closer to the upper and lower edges of the workspace, finally forming a closed loop. The displacement curves of the end-effector in x, y, and z directions can be observed in the Adams/PostProcessor module and exported, as shown in Figure 11, which can reflect the motion of the end-effector in this industrial robot.

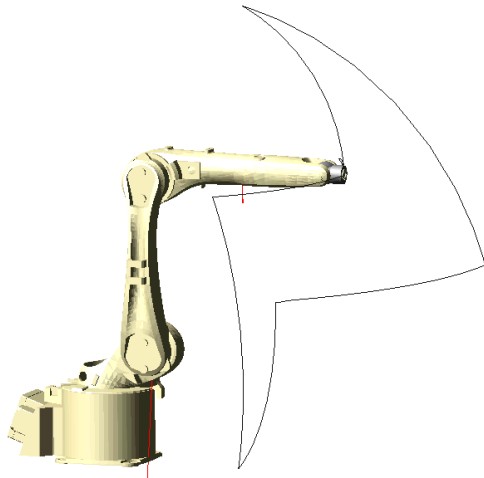

**Figure 10.** Simulation trajectory of robot end-effector.

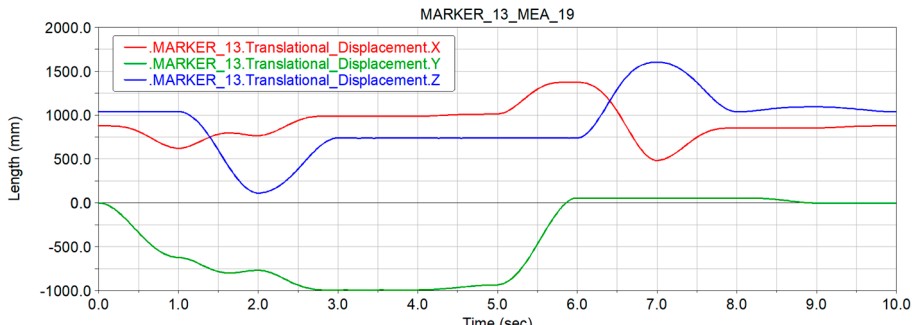

**Figure 11.** Simulation displacement curve of robot in each direction.

It can be seen that the red curve, green curve, and blue curve indicate the displacement values of the end-effector in the x, y, and z directions with time, respectively, and the output values corresponding to these curves can be used as reference values for motion reliability analysis. The displacement in the x-direction has a large change from the 6th to the 7th second, and the end-effector moves away from the center of the robot and approaches the edge of the maximum working space at the 7th second; in the y-direction, the end-effector moves in the negative direction and gradually moves away from the center of the robot, and starts to move in the positive direction at the 5th second and stabilizes after reaching

the origin; in the z-direction, the displacement decreases from the 1st to the 2nd second, and the end-effector approaches the bottom edge, followed by upward dis-placement along the *z*-axis direction.

Figures 12–14 shows the comparison of the simulation results in x, y, and z directions in the ideal state and under the effect of six joint clearances. The local magnification shows that the position of the end-effector is shifted relative to the ideal state, and it is more obvious at the corners. Based on the output motion parameters of the industrial robot end-effector in the ideal state, the deviation values of the output motion parameters under the effect of the six joint clearances can be calculated.

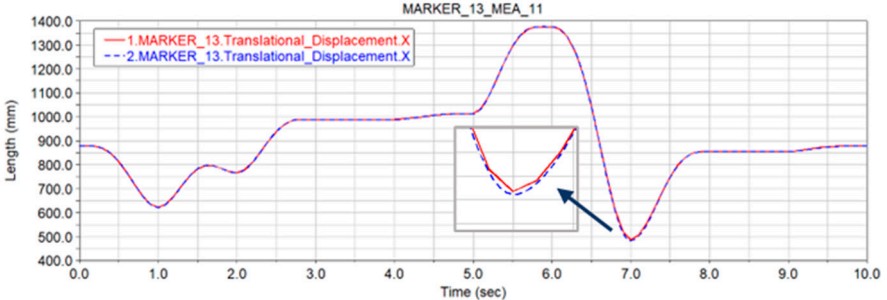

**Figure 12.** Comparison of simulation displacement curves in the x-direction.

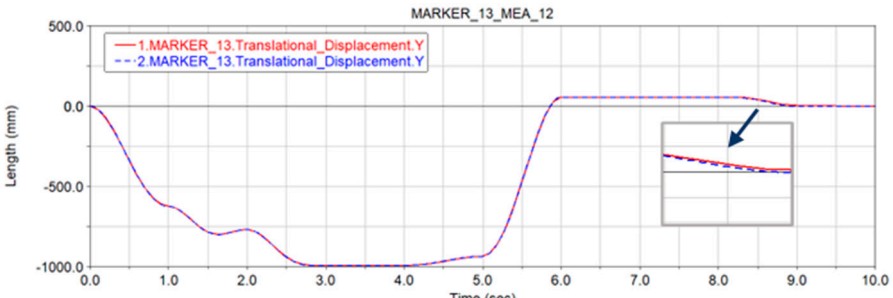

**Figure 13.** Comparison of simulation displacement curves in the y-direction.

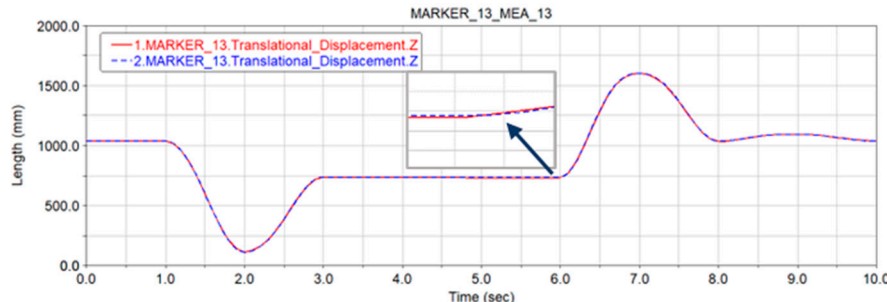

**Figure 14.** Comparison of simulation displacement curves in the z-direction.

The position error interval of the robot end-effector in the x-direction shown in Figure 15 is [−3.0183, 0.6975] mm, and the allowable accuracy of the position error of the robot end-effector is [−3,3] mm. From the robot end-effector position error curve, it can be seen that the position error of point 70 is the largest, and the error value is −3.0183 mm. The position error interval of the y-direction of the robot end-effector shown in Figure 16 is [−3.2367, 1.8808] mm, and the allowable accuracy of the position error of the robot end-effector is [−3,3] mm. From the robot end-effector position error curve, it can be seen that the attitude error of point 91 is the largest, and the error value is −3.2367 mm. The position error interval of the z-direction of the robot end-effector shown in Figure 17 is [−0.7244, 3.0768] mm, and the allowable accuracy of the position error of the robot end-effector is

[−3,3] mm. From the robot end-effector position error curve, it can be seen that the attitude error of point 53 is the largest, and the error value is 3.0768 mm.

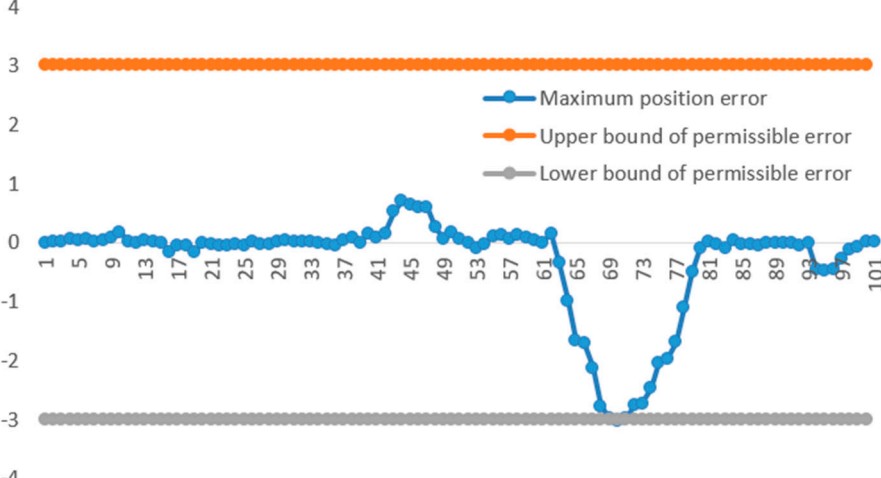

**Figure 15.** Position error of each trajectory point in the x-direction.

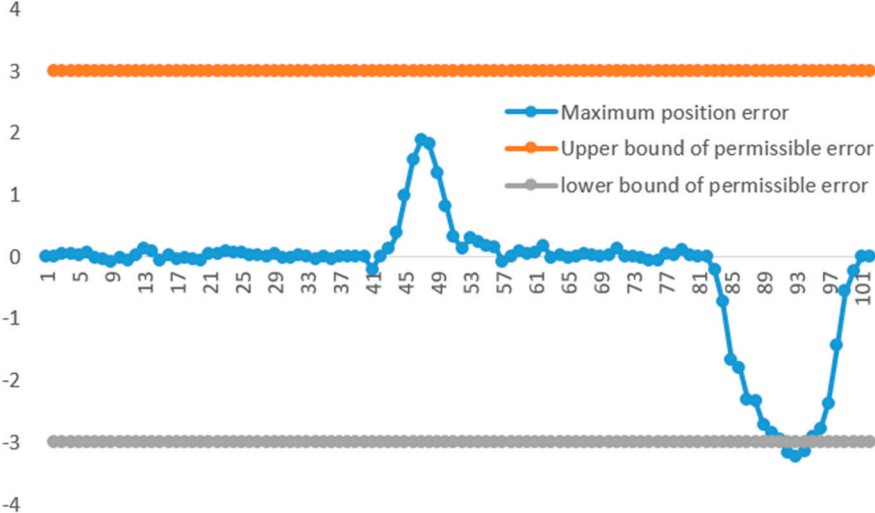

**Figure 16.** Position error of each trajectory point in the y-direction.

Combining with Figures 12–14, Figures 15–17, it can be seen that: (1) most of the position errors of the end-effector in x, y, and z directions occur at the inflection points of the displacement curve, which is due to the large change of velocity at the inflection points of the displacement curve, and the clearance is more sensitive to the velocity change of the mechanism, making the mechanism produce position errors. For example, the sudden change of position error at points 60–80 in Figure 15 is due to the continuous turn of displacement of the end-effector at 6–8s in the x-direction, which means that the speed of the end-effector changes a lot in this time period, thus leading to a large position error; (2) combined with the motion trajectory of the end-effector, it is found that when a large position error occurs, the end-effector moves to the edge of the workspace, and when it is close to the center of the workspace, the position error can be controlled within a small range. For example, in Figure 17, the position error gradually increases from point 46 and reaches the maximum position error around point 53, at which time the end-effector is found to move to the farthest edge of the front side of the robot workspace, and the position error starts to decrease with the subsequent move away from the edge. Therefore, it can be

inferred that the reliability of the end-effector is related to its position within the workspace, and the further it is from the center of the workspace, the less reliable it is.

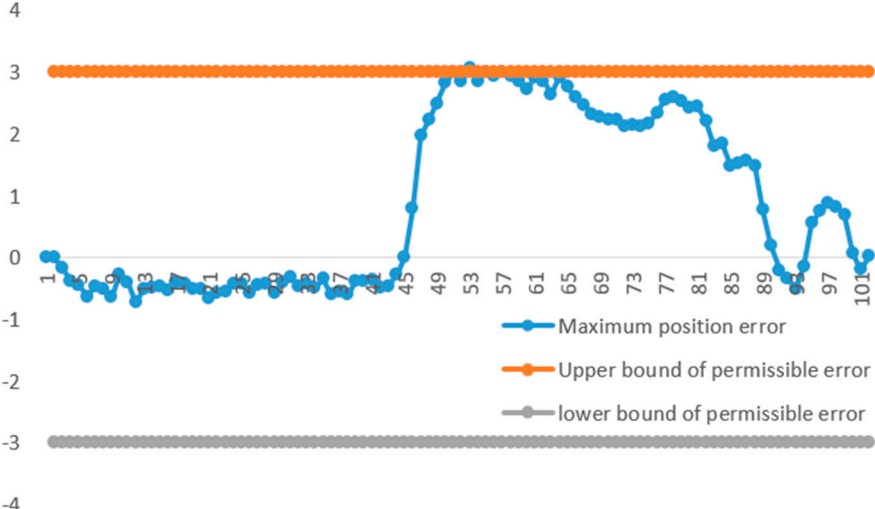

**Figure 17.** Position error of each trajectory point in the z-direction.

According to the mechanism motion reliability analysis method proposed in Chapter 2, the robot motion reliability is obtained by taking the displacement allowable output limit error value as [−3,3]. The probabilistic motion reliability of the robot is obtained by simulation with the Monte Carlo method, and the reliability is calculated by referring to the structural reliability analysis method proposed [28], and the comparison results are shown in Figure 18.

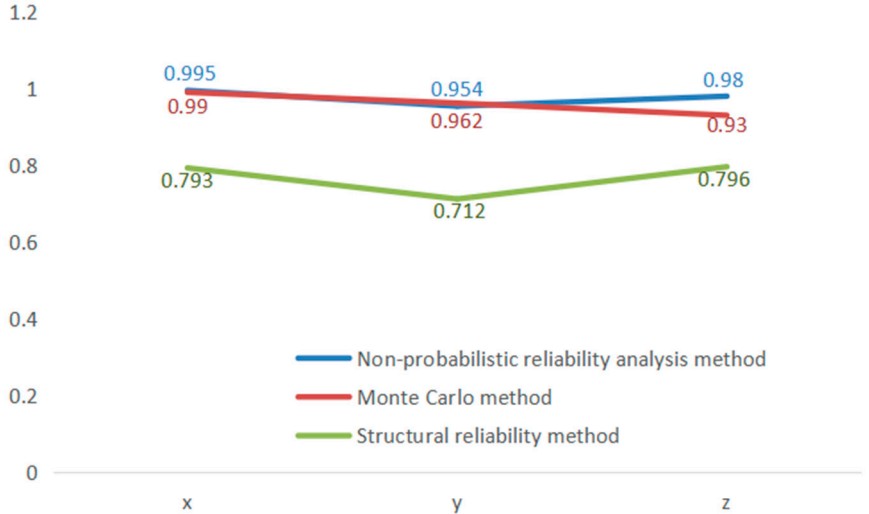

**Figure 18.** Motion reliability of robot end effector.

As seen in Figure 18, the motion reliability in the y-direction is the lowest at 0.954, and the motion reliability in the x-direction is the highest at 0.995. The motion reliability of the robot obtained by the non-probabilistic motion reliability analysis method proposed in this paper is basically consistent with that obtained by simulation with the Monte Carlo method, while the overall reliability is higher than the structural reliability analysis method based on interval theory in the literature. It not only shows the correctness of the non-probabilistic motion reliability analysis method proposed in this paper, but also solves the problem of small values of calculation results and conservative results com-pared with the interval motion reliability analysis method in the previous paper, and is closer to

the real probability. Moreover, with the increase in positioning accuracy required for the end-effector, the overall reliability is bound to decrease significantly, indicating that the motion of the robot is significantly affected by the six joint clearances.

### 4.3. Analysis of Different Time Periods of Robot End-Effectors

The displacement errors generated by the robot motion were divided by time periods, and every 2 s was used as a period to derive the robot motion displacement error range for each time period in Table 5.

**Table 5.** The range of displacement error for each time period of robot simulation motion.

| Time Period | X | Y | Z |
|---|---|---|---|
| 0–2 s | [−0.1667,0.1726] | [−0.1056,0.1183] | [−0.7244,0] |
| 2–4 s | [−0.0255,0.1218] | [−0.2035,0.0717] | [−0.5962,−0.3404] |
| 4–6 s | [−0.0826,0.6975] | [−0.0893,1.8808] | [−0.4825,3.0768] |
| 6–8 s | [−3.0183,0.1457] | [−0.0601,0.1421] | [2.121,2.9081] |
| 8–10 s | [−0.4716,0.0345] | [−3.2367,0.0064] | [−0.4892,1.8582] |

Figure 19 shows that the robot end-effector travels through different workspaces at different times and more comprehensively encompasses the sides of the workspace, the lower and upper parts of the positive direction, and areas with different distances from the center of the workspace.

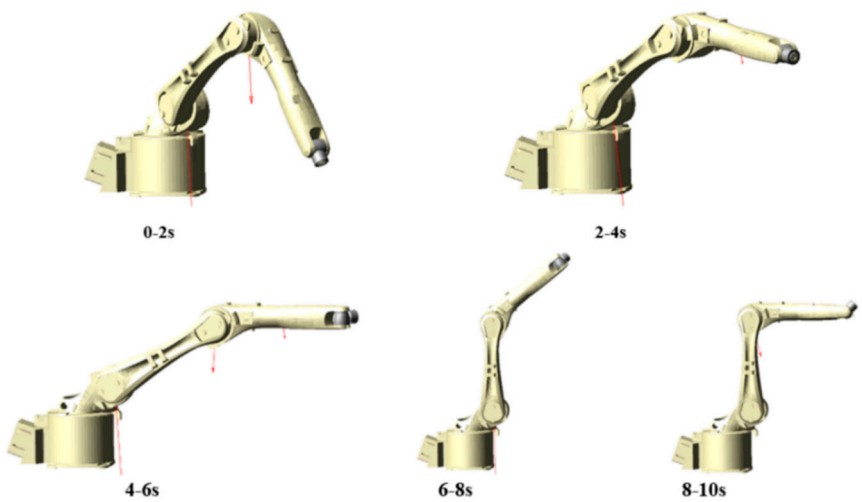

**Figure 19.** Different working spaces traveled by end-effectors at different times.

Figure 20 shows the motion reliability of the robot in each time period, and there is a large difference in motion reliability at different time periods. It can be seen that the motion reliability in the x-direction decreases in 6-8s, and the error varies greatly, with the highest offset of 3.0183 mm from the ideal trajectory. The reliability of the motion in the y-direction decreases in 8–10 s, only 0.927, and the displacement error changes greatly; the reliability of the displacement in the z-direction is lower in 4–6 s when the end-effector is far from the center of the working space, and the displacement fluctuates significantly due to the influence of joint clearance as well as the contact force; while in the time period of 0-4s, the reliability of the displacement in the x, y and z directions is 1, the motion of the robot end effector is completely reliable.

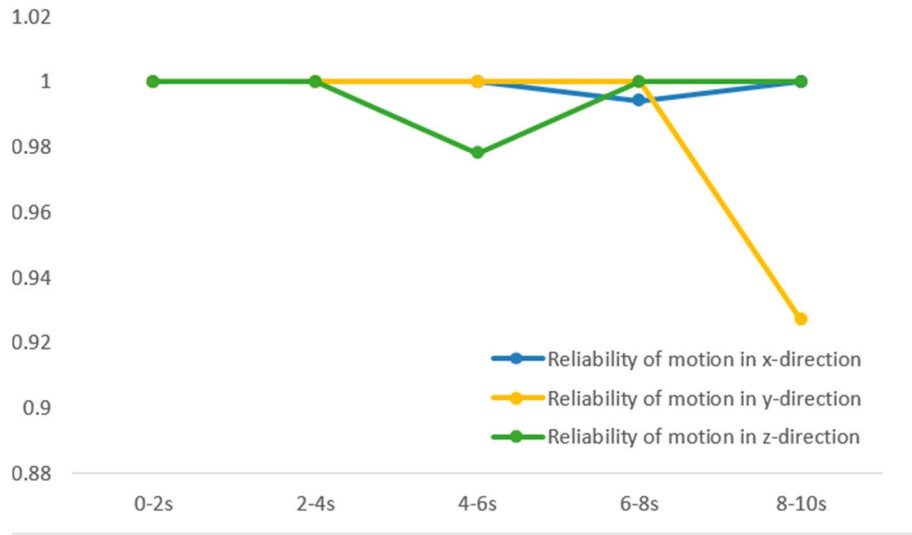

**Figure 20.** Non-probabilistic reliability of robot motion at different time periods.

This shows that the influence of uncertain joint clearance parameters on the motion reliability of the robot end-effector varies with different workspace regions, and the reliability of positioning accuracy is different in different spatial regions. The movement reliability of the end effector is high in the less affected workspace area, and with the change in the workspace area, the influence degree of the uncertain joint clearance parameters will also change, and the movement reliability of the end effector will rise or fall.

## 5. Conclusions

In this paper, the reliability measure of the robot end-effector is studied based on the non-probability interval theory. The position permit interval and error interval of the end-effector are specified. According to the relationship between the size of the upper and lower bound of the interval, and combined with the two-dimensional coordinate system, the different states of reliability can be judged intuitively. For the state in the unstable state, the law analysis is carried out, the concept of non-probability reliability of robot position accuracy is defined, the physical meaning is given, and the positioning accuracy reliability assessment model is established. The principle of the influence of joint clearance on positioning accuracy is analyzed, and it can be seen that the positioning accuracy error of the end-effector is accumulated from the joint clearance error of the full motion pair, and it is believed that the size of the joint clearance error has a direct influence on the reliability of the end-effector. If the degree of joint clearance offset of the motion pair in different workspace areas is different, then the positioning accuracy reliability of the robot end-effector in different workspace areas is also different. The motion path is split into five segments by time, and the workspace area for each segment of end-effector motion is different, and the reliability of each segment's motion is analyzed. The main conclusions are as follows:

(1) Non-probabilistic reliability is introduced to analyze the robot position accuracy under the influence of uncertainty considering the joint clearance, and it is found that the method in this paper is consistent with the traditional probabilistic method in evaluating the reliability of the robot end position, but the method in this paper has a lower degree of data requirement and has obvious advantages for the reliability analysis of small sample and information-poor structured systems.

(2) The simulation analysis of the robot under the influence of the ideal state and uncertain joint clearance parameters is carried out to obtain the displacement error range in each spatial direction, and the reliability analysis based on the non-probability interval theory is carried out. The reliability of the end-effector of the robot is related to the displacement curve. The reliability decreases at the inflection point of the displace-

ment curve and is higher at the smooth displacement curve, and the phenomenon of uneven transition of the displacement curve should be avoided when planning the robot motion trajectory; the reliability of the end-effector is related to its position in the workspace, and the further away from the center of the workspace, the worse the reliability.

(3) The motion path is divided into small segments in time, and the reliability of each segment's motion is analyzed to compare the reliability of the end-effector in different workspace areas corresponding to different time segments, and the conclusion is drawn that the reliability of the end-effector changes in different working regions under the influence of uncertain joint clearance parameters. Based on this conclusion, the research direction of dividing the robot space and partitioning it to establish a non-probability-based reliability calibration model is proposed, which can realize the prediction of the robot end-effector position range and the interval identification of parameters and improve the positioning accuracy and calibration reliability of the robot in the full workspace region.

**Author Contributions:** Conceptualization, methodology, study design, Z.T. and J.P.; writing—original draft preparation, J.P.; writing—review and editing, J.S.; software, data collection and analysis, simulated analysis, validation, J.S. and X.M. All authors have read and agreed to the published version of the manuscript.

**Funding:** This work was supported by the National Natural Science Foundation of China (51965017), the Key Research and Development Project of Jiangxi Science and Technology Department (20192BBE50006), and the Natural Science Foundation Project of Jiangxi Province (20202BABL204037).

**Institutional Review Board Statement:** Not applicable.

**Informed Consent Statement:** Not applicable.

**Data Availability Statement:** The data used to support the findings of this study are available from the corresponding author upon request.

**Conflicts of Interest:** The authors declare no conflict of interest.

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
