# Peer review of "Non-Probabilistic Reliability Analysis of Robot Accuracy under Uncertain Joint Clearance"

_machines, doi:10.3390/machines10100917_

Round 1

Reviewer 1 Report

The article present an extensive study on the precision metrics and results of a robotic manipulator motion and positioning. 
It presents understandable and replicable methods for analyzing the reliability of the motion and the positioning. 
The references are adequate in the context and number. 
The introduction presents a good state-of-the-art.
The methodology and results are clear and well stated and the conclusions are concise and straight to the point. 
There are some minor typos that can be improved, so the authors must read the manuscript carefully and improve it.
In my opinion, this paper can be accepted and published. 

Reviewer 2 Report

See attached file

Reviewer 3 Report

This paper investigates the motion accuracy of a 6-joint industrial robot end effector based on Adams. Using the D-H (Denavit-Hartenberg) parametric modeling method, the kinematic model of the industrial robot is established, the kinematic equations of the robot are derived and the positive kinematic solution is performed; the end position error model of the industrial robot and the robot motion reliability evaluation model are established based on the non-probability interval theory; combined with the joint clearance modeling theory, the end effector under the influence of 6-joint clearance. I think the proposed work is interesting and some comments are listed as follows.

1.       Please correct and check all possible spelling and grammar mistakes carefully. There are a lot of grammatical mistakes.

2.       The proposed work has already been conducted by several other researchers. Thus it is not clear what is the unique contribution.

3.       The authors should express the contributions in a comparative way. This means that the authors should compare the superiority of their design with respect to the previous published papers in Section 1.

4.       It is suggested to merge the merits listed in the Introduction part and further introduce the design difficulties of the proposed method, rather than only giving the obtained results.

5.       The author has provided several references on the solution to the under-study problem. How the author’s work is different from the other works referred to in the literature by the author, what were the shortcomings of these research works that the author’s proposed work has mitigated.

6.       The literature survey is quite good. However, the discussion from introduction can be improved by adding the following references. For example, “10.1109/ACCESS.2021.3139041” and etc.

7.       The presentation of the methodology is not clear and must be carefully rewritten.

8.       The results discussion section is rather short and could elaborate more on the results per se presented in the work.

9.       Authors could provide a link for the design programs and simulation schemes to clarify this subject and the validation as well.

Round 2

Reviewer 2 Report

Several issues persist:

1) According to Eq (7), g(X) is de difference between two interval quantities (R^I and R^I), M=R^I-R^I. Nevertheless, M seems to be a real quantity, not an interval. Why?

2) X from g(X) was not defined (see Eq (7)). X should be the design variable.

3) Please check Fig. 3. According to Eq (7), [M] should be an interval function. Fig 3 shows M=g(x) as a real function. Why? The failure limit function should also be an interval.

4) It is not evident to understand the meaning of interval reliability. What does this measure express? For example, it is possible to affirm that reliability expresses a measure that a probability will not occur by considering reliability based on a probabilistic definition. Nevertheless, what is the interpretation of the interval reliability obtained in Eq (10). This definition is arbitrary, and it lacks mathematical background. Additional definitions and assumptions should be considered.

5. The positioning error of the end-effector was computed for the simulation trajectory of fig. 9. The interval clearances on the joints produce these errors. Nevertheless, the positioning error was obtained for a single case study (for an arbitrary trajectory); a method to obtain the interval positioning error was not illustrated. Thus, it is impossible to obtain a comprehensive reliability assessment based on the presented results; i. e., the total interval error was not computed for a single manipulator’s pose.

6. The results are inconclusive: the proposed method to evaluate the kinematic reliability was not compared to previous approaches reported in the literature. Moreover, it is not evident to understand the physical meaning and mathematical background of the results.

Round 3

Reviewer 2 Report

Theoretical and methodological mistakes were not corrected:

1. g(x) is a real function in Fig. 4; this definition is not in agreement with the definition of fig. 3. Please check: Moore, R. E., Kearfott, R. B., & Cloud, M. J. (2009). Introduction to interval analysis. Society for Industrial and Applied Mathematics. (chapter 5: Introduction to interval functions.)

2. It is recommended that authors consult: Kang, R., Zhang, Q., Zeng, Z., Zio, E., & Li, X. (2016). Measuring reliability under epistemic uncertainty: Review on non-probabilistic reliability metrics. Chinese Journal of Aeronautics29(3), 571-579. This excellent paper brings light to the theoretical background of non-probabilistic methods.

4. The “motion reliability” lack of mathematical basis. This result does not express an assessment of kinematic reliability as presented in the literature.

3. Results: It is impossible to compare a probabilistic reliability index with a non-probabilistic index, as presented in the results. The correct procedure could be compared to non-probabilistic methods.

Therefore, this paper is unsuitable for publication as an original research paper.
